# Utility of $SpO_2/FiO_2$ ratio for acute hypoxemic respiratory failure with bilateral opacities in the ICU

**Yosuke Fukuda**[1]*, **Akihiko Tanaka**[1], **Tetsuya Homma**[1], **Keisuke Kaneko**[1], **Tomoki Uno**[1], **Akiko Fujiwara**[1], **Yoshitaka Uchida**[1], **Shintaro Suzuki**[1], **Toru Kotani**[2], **Hironori Sagara**[1]

**1** Department of Medicine, Division of Respiratory Medicine and Allergology, Showa University School of Medicine, Shinagawa-ku, Tokyo, Japan, **2** Department of Intensive Care Medicine, Showa University School of Medicine, Shinagawa-ku, Tokyo, Japan

* y.f.0423@med.showa-u.ac.jp

**Data Availability Statement:** All relevant data are within the manuscript and its Supporting Information files.

## Abstract

Acute hypoxemic respiratory failure (AHRF) with bilateral opacities causes fatalities in the intensive care unit (ICU). It is often difficult to identify the causes of AHRF at the time of admission. The $SpO_2$ to $FiO_2$ (S/F) ratio has been recently used as a non-invasive and alternative marker of the $PaO_2/FiO_2$ (P/F) ratio in acute respiratory failure. This retrospective cohort study was conducted from October 2010 to March 2019 at the Showa University Hospital, Tokyo, Japan. We enrolled 94 AHRF patients who had bilateral opacities and received mechanical ventilation in ICU to investigate their prognostic markers including S/F ratio. Significant differences were observed for APACHE II, S/F ratio, $PaO_2/FiO_2$ (P/F) ratio, and ventilator−free-days at day 28 for ICU mortality, and for age, S/F ratio, P/F ratio, duration of mechanical ventilation, and ventilator−free days at day 28 for hospital mortality. Multivariate logistic regression analysis showed that the S/F ratio was significantly and independently associated with the risk of death during in ICU (p = 0.003) and hospitalization (p = 0.002). The area under the receiver operating characteristic curves (AUC) based on the S/F ratio were significantly greater than those based on simplified acute physiology score (SAPS) II and sequential organ failure assessment (SOFA) for ICU mortality (0.785 in S/F ratio vs. 0.575 in SAPS II, p = 0.012; 0.785 in S/F ratio vs 0.594 in SOFA, p = 0.021) and for hospital mortality (0.701 in S/F ratio vs. 0.502 in SAPS II, p = 0.012; 0.701 in S/F ratio vs. 0.503 in SOFA, p = 0.005). In the subanalysis for bacterial pneumonia and interstitial lung disease groups, the AUC based on the S/F ratio was the greatest among all prognostic markers, including APACHE II, SAPS II, and SOFA. The S/F ratio may be a useful and noninvasive predictive prognostic marker for acute hypoxemic respiratory failure with bilateral opacities in the ICU.

## Introduction

Acute hypoxemic respiratory failure (AHRF) documented by the presence of bilateral opacities on X-ray or computer tomography (CT) is a life-threatening condition in the intensive care

**Funding:** The authors received no specific funding for this work.

**Competing interests:** The authors have declared that no competing interests exist.

unit (ICU) [1]. Acute respiratory distress syndrome (ARDS) is a representative condition of AHRF with bilateral opacities that accounts for almost 10% of all ICU admissions and 23% of cases with mechanical ventilation [2]. However, it is often challenging to differentiate ARDS from non-ARDS on admission despite medical history interviews, physical findings, and examinations. In the LUNG SAFE study [2], a multinational prospective cohort study of ARDS, only 60% of ARDS cases were detected by clinicians [2]. They reported that only 34% of ARDS patients who met the Berlin ARDS criteria on admission were identified [2]. Although it is often difficult to define the patient's prognosis owing to the challenge in recognizing and making a differential diagnosis of ARDS on admission, it is crucial to identify prognostic predictors to develop a treatment strategy in patients with the mechanically ventilated AHRF.

We have some tools for the prediction of the patient clinical outcomes with AHRF in the ICU [3]. The Acute Physiology and Chronic Health Evaluation (APACHE) II score was developed in 1985 and constitutes a scoring system that is used to evaluate the severity of illness in the ICU [4], and it is significantly associated with mortality in patients with acute lung injury/ARDS patients [5]. Kao et al. reported that APACHE II is a useful tool for predicting the 60-day mortality in cases of influenza pneumonia-related ARDS [6]. Similarly, the simplified acute physiology score (SAPS) II and sequential organ failure assessment (SOFA)—developed to assess organ failure in the intensive care unit [3]—have also been reported to be useful prognostic indicators in acute respiratory failure [7, 8]. Moreover, pulse oximetry saturation is continually monitored in the ICU as a noninvasive tool for assessing patients' respiratory status. The PaO$_2$/FiO$_2$ (P/F) ratio that defines the severity of ARDS according to the Berlin criteria [9] has an established correlation with the SpO$_2$/FiO$_2$ (S/F) ratio [10, 11]. It was suggested that the S/F ratio may be useful as an alternative marker for the P/F ratio in ARDS [7, 12]. However, the indicators that are useful concerning in-hospital and ICU deaths have been debated. Specifically, now that coronavirus disease-2019 (COVID-19) is rampant worldwide, a simple index that can be monitored continually is needed, even in clinical settings that are resource-limited [13].

This study aimed to investigate which indicators would be useful for patients with the mechanically ventilated AHRF with bilateral opacities in the ICU.

## Materials and methods

This retrospective cohort study was conducted from October 2010 to March 2019 at Showa University Hospital, Tokyo, Japan. The eligibility criteria included age $\geq$ 18 years, use of mechanical ventilation (positive end-expiratory pressure (PEEP) exceeding 5 cmH$_2$O), bilateral opacities on chest X-ray or CT on ICU admission, and an examination of arterial blood gas analysis 24 h after admission. Bilateral opacities on chest X-ray or CT were independently judged by two respiratory specialists according to the Berlin definition of ARDS [9]. When there was disagreement between the specialists on the decision, they discussed it and reached a consensus. Patient characteristics and data, including age, sex, body mass index (BMI), smoking history, length of stay, duration of mechanical ventilation, vital signs, fluid intake/output balance, chest radiographic findings, blood examination findings, and comorbidities, were obtained from the medical records of the patients. Additionally, we evaluated the severity of disease using APACHE II, SAPS II, SOFA, the P/F ratio, and S/F ratio at 24 h after ICU admission [14]. We followed up the patients up to 60 days after admission. The settings of mechanical ventilation were adjusted the FiO$_2$ to maintain SpO$_2$ at 90–95% and PEEP exceeding 5cmH$_2$O according to ARDSNet mechanical ventilation strategy [15] by each physician in charge.

All data were not fully anonymized before we accessed them. The date range during which patients' medical records/samples were assessed was April 2019 to December 2019. After publishing a notice that stating that the research would be based on the patient's clinical information on the website of the Showa University Ethics Committee, we obtained informed consent in the form of opt-out. The Showa University Ethics Committee approved this study and the opt-out consent mechanism (approval number: 2795). The study complied with the principles of the Declaration of Helsinki.

Statistical analysis was performed using JMP (Version15, SAS Institute, Minato-ku, Tokyo, Japan). All data are presented as median (range), or number (percentage), as required. Differences between categorical variables were analyzed using the Fisher's exact test with 95% confidential interval (CI) statistics and continuous variables were analyzed using the Mann–Whitney U test. Receiver operating characteristic (ROC) curves were plotted to estimate the cutoff values for diagnosis and we conducted the likelihood ratio test for goodness of fit. P values less than 0.05 were considered statistically significant.

## Results

A total of 572 patients admitted to ICU were screened (Fig 1). According to the exclusion criteria, we excluded the following patients: 224 patients who did not receive invasive mechanical ventilation, 168 patients who did not have bilateral opacities on X-ray or CT, 50 patients who did not have P/F ratio results at 24 h post-admission, eight patients who died within 24 h of admission to the ICU, two patients who had acute heart failure, and two patients who had an unknown outcome. We included only the first admission to the ICU of the same patient during the study period. Finally, 94 patients were enrolled in this study (Fig 1).

Patient demographic information was presented in Table 1. The median patient age was 72 (62–80) years, and 75.5% were male. The mean APACHE II, SAPS II, and SOFA scores were 29 (24–35.8), 62 (52.3–75), and 14 (11.3–15), respectively. The median S/F and P/F ratios were 192 (152–245), and 180 (121–236). The S/F ratio and the P/F ratio were significantly correlated (Fig 2. S/F ratio = 60.9 + 0.75 × P/F ratio (p < 0.001, r = 0.87)). The disease profile of the 94 patients was as follows: bacterial pneumonia accounted for 38 patients, interstitial lung diseases (ILD) for 36, viral or pneumocystis pneumonia for nine, aspiration pneumonia for four, and others for seven. All patients with bacterial pneumonia, viral pneumonia, and pneumocystis pneumonia were septic.

Table 2 showed differences between survivors and nonsurvivors in the ICU and hospital. Significant differences were observed for APACHE II, S/F ratio, P/F ratio, and ventilator–free days at day 28 for ICU mortality (p = 0.027, p < 0.001, p < 0.001, p < 0.001, respectively), and for age, S/F ratio, P/F ratio, length of mechanical ventilation, and ventilator–free days at day 28 for hospital mortality (p = 0.002, p = 0.001, p = 0.002, p < 0.001, p < 0.001, respectively).

We used multivariate logistic regression models to determine the risk factors associated with death in the ICU and hospital (Table 3). While only the S/F ratio was independently associated with mortality in the ICU (p = 0.002), age and the S/F ratio were significant variables associated with mortality in the hospital (p = 0.001, p = 0.002, respectively).

The area under the ROC curves (AUCs) for hospital mortality are presented in Fig 3. The AUC of the S/F ratio was 0.785 (95% CI 0.67–0.899, p = 0.002) for ICU mortality and 0.701 (95% CI 0.59–0.813, p = 0.002) for hospital mortality. The AUC of the S/F ratio was significantly greater than that of SAPS II and the SOFA for ICU mortality (S/F ratio vs. SAPS II, p = 0.012; S/F ratio vs. SOFA, p = 0.021) and hospital mortality (S/F ratio vs SAPS II, p = 0.012; S/F ratio vs. SOFA, p = 0.005). Subgroup analysis of patients with bacterial pneumonia or ILD also showed that the S/F ratio had the highest AUC among the four prognostic

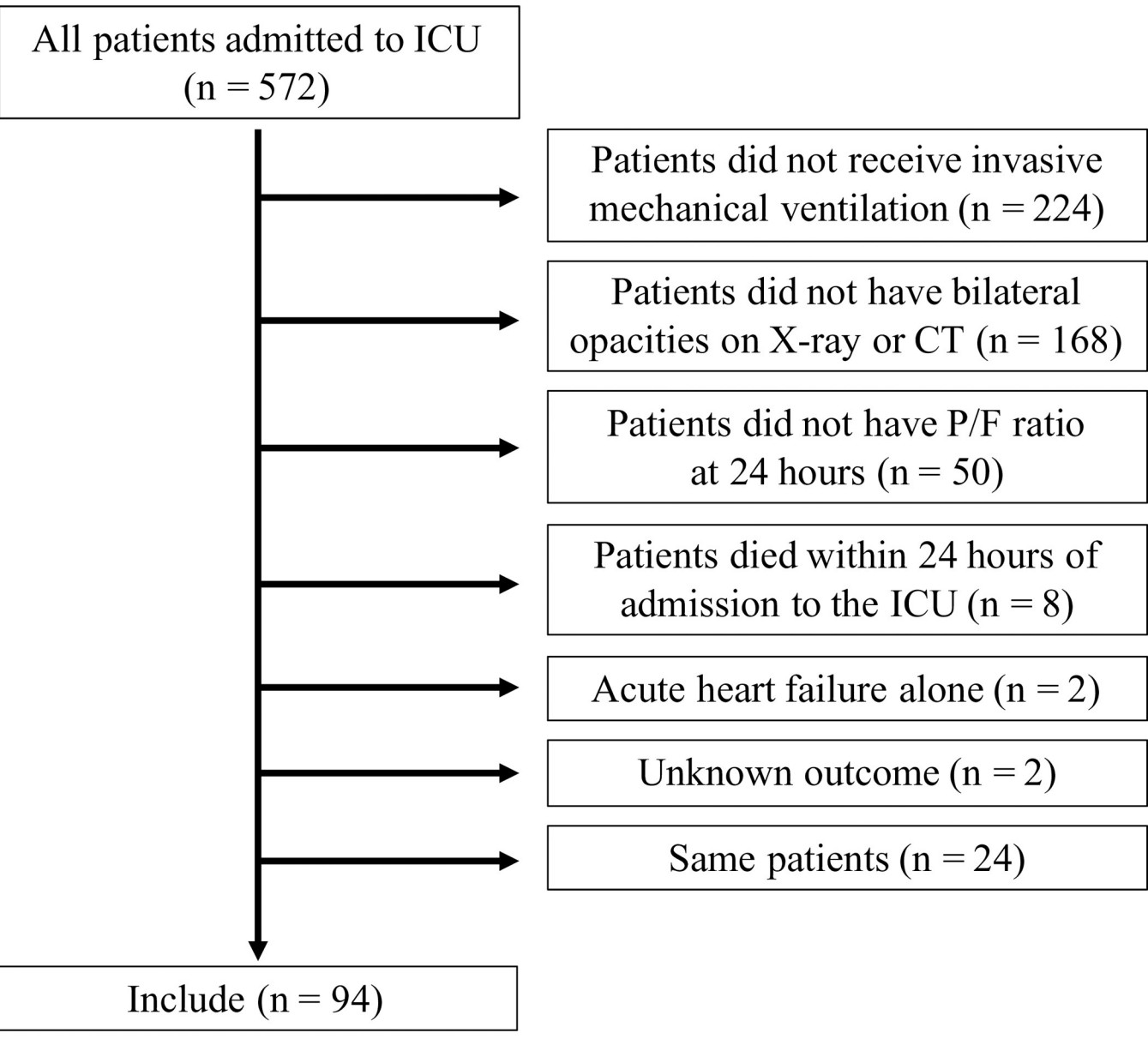

**Fig 1. The study flow diagram in this study.** 572 patients in intensive care unit (ICU) were screened. After excluding the patients who met the exclusion criteria, 94 patients were enrolled in current study.

factors (Fig 4A–4D: bacterial pneumonia, 0.971 (95% CI 0.907–1.000, p = 0.001) for ICU mortality; Fig 4A: 0.700 (95% CI 0.493–0.908, p = 0.046) for hospital mortality; Fig 4B: interstitial lung disease, 0.697 (95% CI 0.520–0.875, p = 0.113) for ICU mortality; Fig 4C: 0.720 (95% CI 0.539–0.902, p = 0.018) for hospital mortality; Fig 4D).

## Discussion

Our findings underscored that S/F ratio was easy to obtain a superior prognostic factor for mechanically ventilated AHRF with bilateral opacities in the ICU to APACHE II, SAPS II, and SOFA.

We showed that the S/F ratio was significantly higher for survivors compared with nonsurvivors in the hospital and ICU. In the pediatric ICU, the S/F ratio was considered a useful and

**Table 1. Clinical characteristics in total patients.**

| | Total patients |
|---|---|
| | **n = 94** |
| Age, year | 72 (62–80) |
| Sex, male/female (% of male) | 71 / 23 (75.5) |
| BMI, kg/m$^2$ | 22.2 (19.6–24.4) |
| APACHE II | 29 (24–35.7) |
| SAPS II | 62 (52.3–75) |
| SOFA | 14 (11.2–15) |
| S/F ratio | 192 (152–245) |
| P/F ratio | 179 (121–236) |
| Use of systemic corticosteroids, No. (%) | 71 (75.6) |
| Use of sivelestat sodium, No. (%) | 29 (30.9) |
| Use of vasopressor, No. (%) | 53 (56.4) |
| Length of stay in hospital, days | 31 (19.2–58.7) |
| Length of stay in ICU, days | 10 (6.3–21.5) |
| Duration of mechanical ventilation, days | 9 (5–24.5) |
| Ventilator—free days at day 28, days | 15.5 (0–22.2) |
| Comorbidities, No. (%) | |
| Hypertension | 45 (47.9) |
| COPD | 28 (29.8) |
| Malignancy | 27 (28.7) |
| Diabetes mellitus | 25 (26.6) |
| Coronary artery disease | 16 (17.1) |
| Chronic kidney disease | 9 (9.6) |
| Autoimmune disease | 9 (9.6) |
| Diagnosis, No. (%) | |
| Bacterial pneumonia | 38 (40.4) |
| Interstitial lung disease | 36 (38.2) |
| Viral/Pneumocystis pneumonia | 9 (9.5) |
| Aspiration pneumonia | 4 (4.2) |
| Others | 7 (10.6) |

Data are presented as median (range), or number (percentage). APACHE acute physiology and chronic health evaluation, BMI body mass index, COPD chronic obstructive pulmonary disease, ICU intensive care unit, SAPS simplified acute physiology score, SOFA sequential organ failure assessment.

noninvasive marker because of difficulty with daily blood gas sampling to calculate the P/F ratio. Wong et al. enrolled 70 children with ARDS and recorded some parameters on days 1, 3, and 7 [16]. They concluded that a low S/F ratio on the day of diagnosis was associated with the number of ventilator-free days and 28 days free of ICU admission. Another report showed that the S/F ratio was a readily available marker for detecting a high risk of death in pediatric patients with acute hypoxemic failure [17]. Conversely, data on whether the S/F ratio predicted mortality in adult patients was limited. Bass et al. focused on the usefulness of combining the S/F ratio and lung ultrasound (LUS) for AHRF patients, particularly ARDS [18]. They reported that the combination of S/F ratio and LUS identified ARDS with a sensitivity of 91% and a specificity of 48% [18]. A study in Rwanda, one of the developing countries with limited health care resources, found that the Kigali modification of the Berlin definition, including the use of the S/F ratio as an alternative to the P/F ratio, was useful for the prediction of the prognosis of

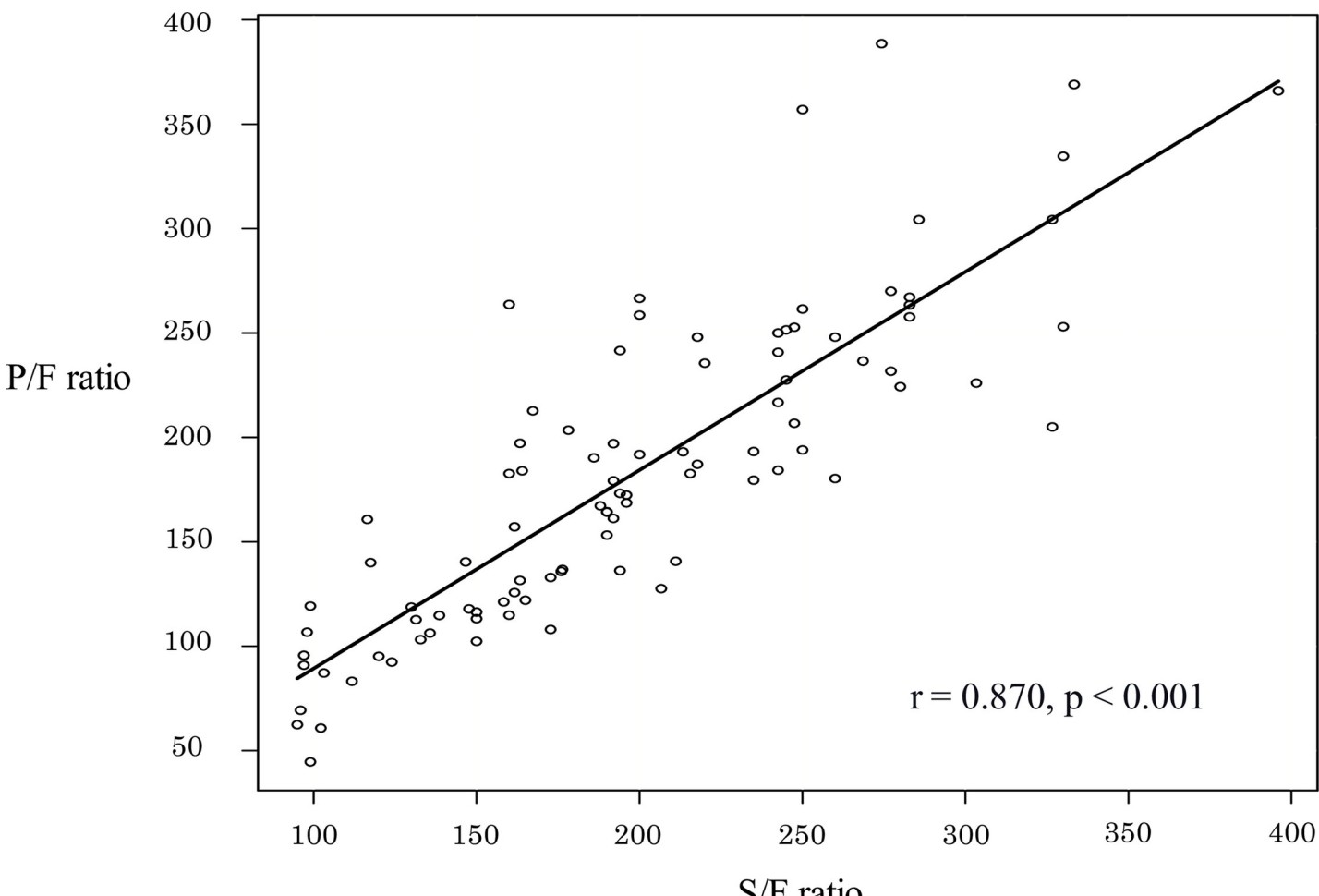

**Fig 2. The relationship between the SpO$_2$/FiO$_2$ (S/F) and PaO$_2$/FiO$_2$ (P/F) ratios.** S/F ratio showed significant linear correlation. S/F ratio = 60.9 + 0.75 × P/F ratio (p < 0.001; r = 0.87).

ARDS patients [19]. Sendagire et al. also showed that the modified SOFA score using the S/F ratio instead of the P/F ratio was useful for the prediction of ICU mortality in a resource-limited setting (e.g., settings in which blood gas analysis data were lacking) [20]. Therefore, we considered that the S/F ratio would be useful markers to predict the mortality of AHRF with bilateral opacities in ICU.

We inferred that the S/F ratio was comparable to other indices that predict the prognosis of AHRF with bilateral opacities. In clinical practice, it is difficult to distinguish between ARDS and acute exacerbation of interstitial lung disease in patients with respiratory failure immediately after admission to the ICU, therefore it is useful to find prognostic factors for AHRF. The S/F ratio is a single indicator, and few studies directly compared its usefulness to the combined indices of APACHE II, SAPS II, and SOFA. In a multicenter cohort study of adult ARDS patients in the United States, both lower S/F ratio and lower APACHE II scores were related to early intubation in the ICU [21]. In another report on ARDS, saturation-based predictors, including the S/F ratio, were independently associated with clinical outcomes, whereas the P/F ratio was not associated with these outcomes [22]. One explanation for this finding is attributed to the fact that saturation-based markers are more sensitive and are continually available for the prediction of hospital mortality owing to ARDS compared with blood sample-based

**Table 2. Clinical characteristics between survivor or non-survivor in the hospital and ICU.**

| | ICU | | | | Hospital | |
| --- | --- | --- | --- | --- | --- | --- |
| | Survivor | Nonsurvivor | p value | Survivor | Nonsurvivor | p value |
| | (n = 75) | (n = 19) | | (n = 58) | (n = 36) | |
| Age, year | 71 (60–79.5) | 76 (66–81.5) | 0.256 | 68 (55.5–79) | 76 (68–84) | 0.002 |
| Sex, male/female (% of male) | 56 / 19 (74.6) | 15 / 4 (78.9) | 1.000 | 42 / 16 (72.4) | 29 / 7 (80.6) | 0.463 |
| BMI, kg/m$^2$ | 22.6 (20.0–24.7) | 21.5 (19.4–22.8) | 0.364 | 22.9 (20.5–25.3) | 21.5 (18.9–23.0) | 0.084 |
| APACHE II | 28 (24–34) | 36 (25.5–40.5) | 0.027 | 27.5 (24–34) | 32 (25.5–38.3) | 0.109 |
| SAPS II | 60 (51.5–74.5) | 66 (56–77.5) | 0.311 | 60.5 (51.5–76) | 64.5 (52.7–74.2) | 0.969 |
| SOFA | 13 (11–15) | 14 (12.5–15) | 0.206 | 13 (11.2–15) | 14 (11.7–15) | 0.972 |
| S/F ratio | 200 (163.3–250) | 132.8 (114–189) | <0.001 | 200 (165.5–250) | 162.5 (117.2–303.3) | 0.001 |
| P/F ratio | 193.2 (136.2–249) | 119.2 (104.9–162.4) | <0.001 | 192.4 (140.4–252.4) | 138.1 (105.8–202.1) | 0.002 |
| Use of systemic corticosteroids, No. (%) | 54 (72) | 17 (89.4) | 0.143 | 41 (70.7) | 30 (83.4) | 0.219 |
| Use of sivelestat sodium, No. (%) | 24 (32) | 5 (26.3) | 0.251 | 15 (29.5) | 14 (32.6) | 0.251 |
| Use of vasopressor, No. (%) | 42 (56) | 11 (57.8) | 1.000 | 33 (56.9) | 20 (55.6) | 1.000 |
| Duration of mechanical ventilation, days | 8 (5–25.5) | 13 (7–24) | 0.398 | 6.5 (4–17) | 17 (11–36) | <0.001 |
| Ventilator—free days at day 28, days | 20 (0–23) | 0 (0–11) | <0.001 | 21.5 (11–24) | 0 (0–13.7) | <0.001 |
| Comorbidities, No. (%) | | | | | | |
| Hypertension | 36 (48) | 9 (47.3) | 1.000 | 29 (50) | 16 (44.5) | 0.673 |
| COPD | 23 (30.6) | 5 (26.3) | 0.786 | 19 (32.7) | 9 (25) | 0.492 |
| Malignancy | 21(28) | 6 (31.5) | 0.781 | 14 (24.2) | 13 (36.2) | 0.246 |
| Diabetes mellitus | 22 (29.3) | 3 (15.7) | 0.383 | 17 (30.4) | 8 (22.3) | 0.483 |
| Coronary artery disease | 14 (18.6) | 2 (10.5) | 0.779 | 9 (15.6) | 7 (19.5) | 0.779 |
| Chronic kidney disease | 6 (8) | 3 (15.7) | 0.380 | 5 (8.7) | 4 (11.1) | 0.728 |
| Autoimmune disease | 8 (10.6) | 1 (5.2) | 0.681 | 7 (12.1) | 2 (5.6) | 0.475 |

Data are presented as median (range), or number (percentage). APACHE acute physiology and chronic health evaluation, BMI body mass index, COPD chronic obstructive pulmonary disease, ICU intensive care unit, SAPS simplified acute physiology score, SOFA sequential organ failure assessment

markers. APACHE II, SAPS II, and SOFA are extremely useful in the ICU, but we believe that the S/F ratio is more intuitive and straightforward than other indices.

We showed the usefulness of the S/F ratio as a predictor of mortality in AHRF patients with bilateral opacities in bacterial pneumonia. Interestingly, while the S/F ratio exhibited excellent accuracy in predicting ICU mortality in bacterial pneumonia, it yielded a fair prediction accuracy of hospital mortality. A previous report showed that the lowest S/F ratio tertile (S/F ratio < 164) at ICU admission was associated with hospital mortality compared with the highest S/F ratio tertile (S/F ratio > 236) in the instances in which patients with severe sepsis and septic shock in the ICU were separated into three groups according to the S/F ratio [23]. Our

**Table 3. Multivariate logistic regression analysis for risk of death in the hospital and ICU.**

| | ICU | | | Hospital | | |
| --- | --- | --- | --- | --- | --- | --- |
| | OR | 95% CI | p value | OR | 95% CI | p value |
| Age | 1.020 | 0.975–1.080 | 0.337 | 1.070 | 1.030–1.120 | 0.001 |
| S/F ratio | 0.982 | 0.970–0.994 | 0.002 | 0.987 | 0.978–0.995 | 0.002 |
| APACHE II | 1.040 | 0.968–1.120 | 0.284 | 1.010 | 0.948–1.080 | 0.762 |

APACHE acute physiology and chronic health evaluation, ICU intensive care unit

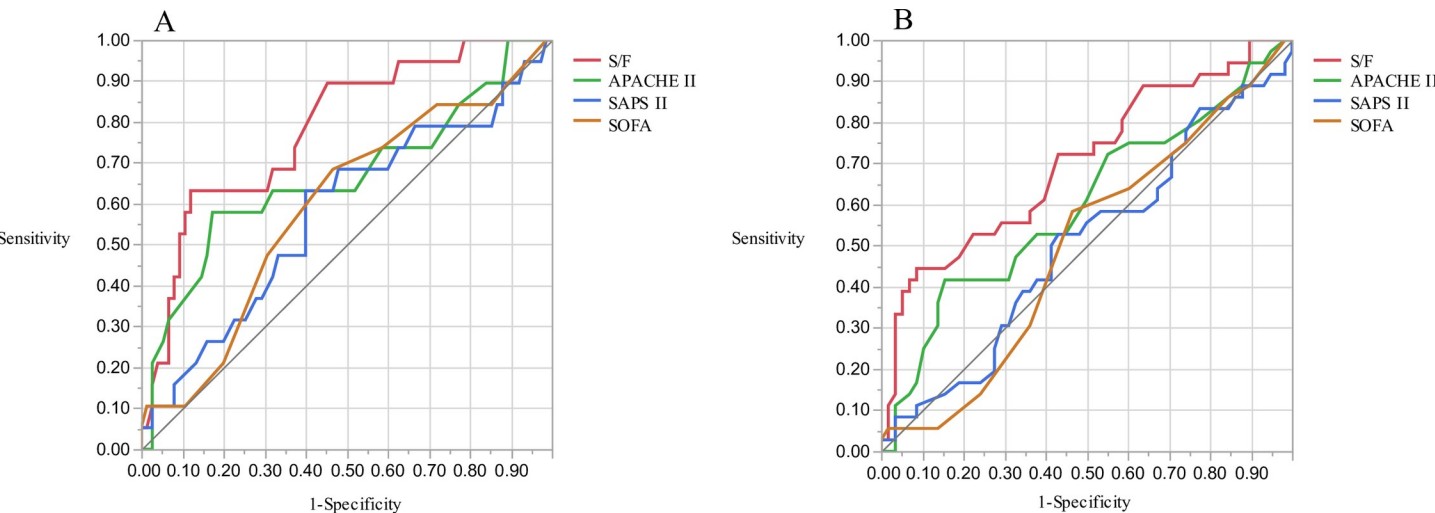

**Fig 3. Receiver Operating Characteristic (ROC) curves for ICU mortality and hospital mortality.** AUC for ICU mortality was 0.784 (A, 95%CI 0.648–0.877) and for hospital mortality was 0.701 (B, 95% confidence interval (CI) 0.579–0.800) using S/F ratio. The AUCs based on the S/F ratio were significantly greater than those based on simplified acute physiology score (SAPS) II and sequential organ failure assessment (SOFA) for ICU mortality (0.785 in S/F ratio vs. 0.575 in SAPS II, p = 0.012; 0.785 in S/F ratio vs 0.594 in SOFA, p = 0.021) and for hospital mortality (0.701 in S/F ratio vs. 0.502 in SAPS II, p = 0.012; 0.701 in S/F ratio vs. 0.503 in SOFA, p = 0.005). Both the cutoff for hospital mortality and the cutoff for ICU mortality were 147.69.

results suggest that the S/F ratio may be a powerful prognostic indicator for both hospital mortality and ICU mortality in bacterial infection-based AHRF with bilateral opacities. In the interstitial pneumonia subgroup, the S/F ratio was a relatively useful measure compared with other measures with low accuracies for hospital and ICU mortality. While low P/F ratio, high positive end-expiratory pressure, age, and low APACHE III score values were associated with hospital mortality in ILD patients that required mechanical ventilation [24], there were no reports that examined the usefulness of the S/F ratio in ILD for hospital and ICU mortality. Our finding was new, but we thought that the S/F ratio's usefulness needs further study because of the poor–fair accuracy.

Our study is associated with several limitations. Firstly, this study was a retrospective cohort study. Because all data were obtained from medical records, selection bias was inherent. Secondly, the number of patients included in this study was not enough because it was a single-center study. We considered that further research is needed to answer the clinical question of whether the S/F ratio is prognostically useful in acute hypoxemic respiratory failure. Thirdly, this study included patients with SpO₂ > 97%. The reason for this is that some patients remained on liberal oxygenation therapy with high SpO₂ control, although we aimed to implement conservative therapy with a target of 90–95% SpO₂ as a mechanical ventilator management [25]. It is known that the oxyhemoglobin dissociation curve is flat in situations where SpO₂ >97%, by contrast, Kwack et al. reported that the S/F ratio in patients with SpO₂ > 97% was inconsistently useful for predicting acute deterioration [26]. Fourthly, we found no evidence that indicated that the S/F ratio at ICU admission is useful. We must be cautious in our interpretation of SpO₂, as FiO₂ is often set high immediately after the start of mechanical ventilator management. Future research is needed to evaluate the S/F ratio's usefulness in earlier phases in the ICU.

## Conclusions

In conclusion, our findings suggest that the S/F ratio may be useful for assessing the impact on clinical outcomes of the mechanically ventilated AHRF with bilateral opacities. When patients

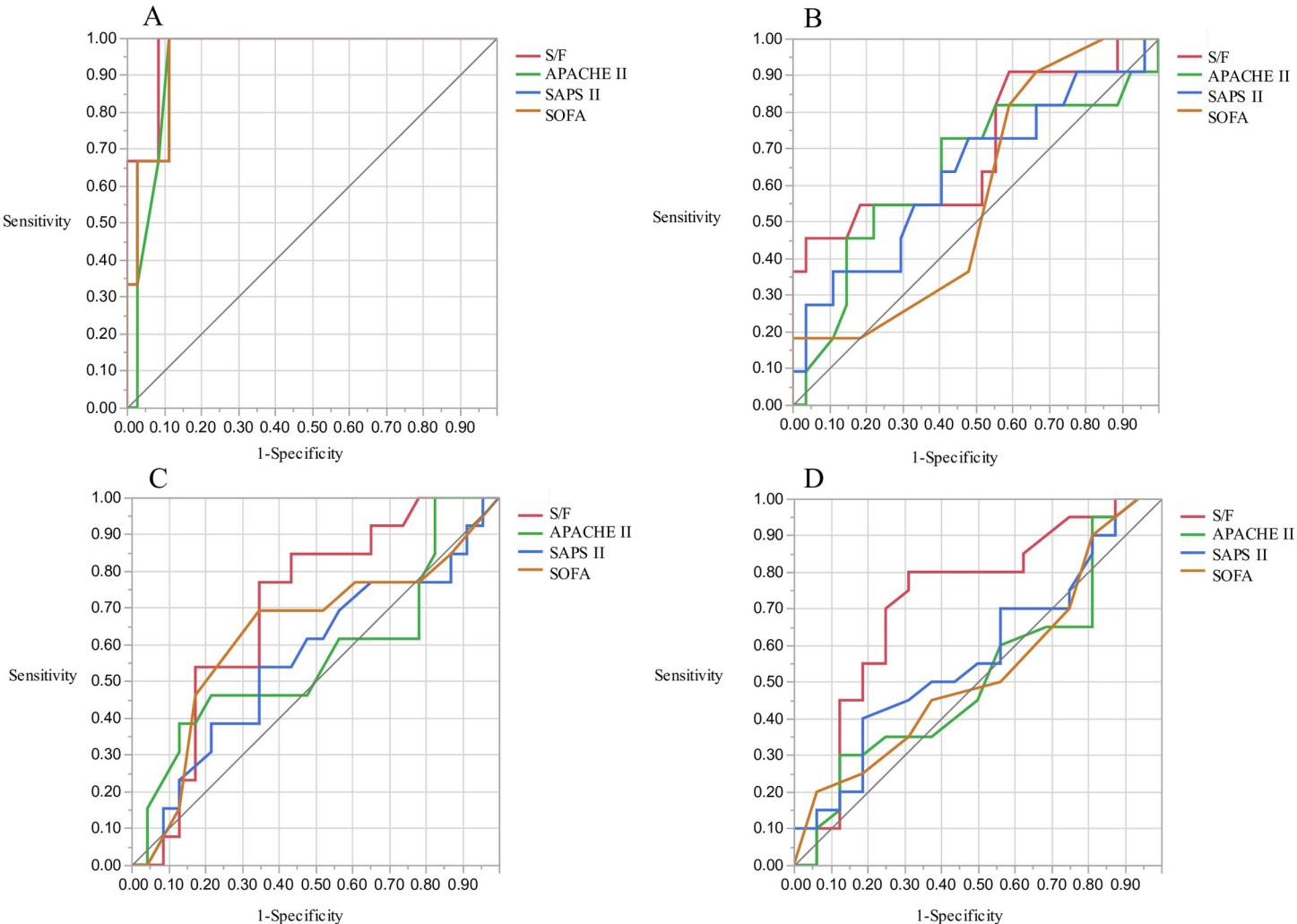

**Fig 4. ROC curves for ICU and hospital mortality in bacterial pneumonia and interstitial lung diseases group.** In the bacterial pneumonia group, the AUC for ICU mortality was 0.971 (A, 95% CI 0.907–1.000) and for hospital mortality was 0.700 (B, 95% CI 0.493–0.908) using the S/F ratio. In interstitial lung diseases group, AUC for ICU mortality was 0.697 (C, 95% CI 0.520–0.875) and for hospital mortality was 0.720 (D, 95% CI 0.539–0.902) using the S/F ratio.

were admitted to the ICU, insufficient information is available. Some formulas exist and are used to predict clinical outcomes for patients, but most of them require scoring methods. Because the S/F ratio does not require complicated calculations, we believe the S/F ratio is simple, can be monitored, and useful for predicting any clinical outcomes in acute hypoxemia.

## Supporting information

**S1 Table. Database of patients characteristics.**
(XLSX)

## Acknowledgments

We gratefully thank the intensive care unit's medical staff at Showa University Hospital for assistance and contributions to our study. We thank enago (www.enago.jp) for English language editing.

## Author Contributions

**Conceptualization:** Yosuke Fukuda.

**Data curation:** Keisuke Kaneko, Akiko Fujiwara, Yoshitaka Uchida.

**Formal analysis:** Keisuke Kaneko, Akiko Fujiwara, Yoshitaka Uchida.

**Supervision:** Akihiko Tanaka, Tetsuya Homma, Toru Kotani.

**Writing – original draft:** Yosuke Fukuda, Tomoki Uno.

**Writing – review & editing:** Akihiko Tanaka, Tetsuya Homma, Shintaro Suzuki, Toru Kotani, Hironori Sagara.

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
