## [Decision Letter · Decision Letter 0]

21 Dec 2020

PONE-D-20-35947

Utility of SpO2/FiO2 ratio for acute hypoxemic respiratory failure with bilateral opacities in the ICU

PLOS ONE

Dear Dr. Yosuke,

Thank you for submitting your manuscript to PLOS ONE. After careful consideration, we feel that it has merit but does not fully meet PLOS ONE’s publication criteria as it currently stands. Therefore, we invite you to submit a revised version of the manuscript that addresses the points raised during the review process.

We look forward to receiving your revised manuscript.

Kind regards,

Aleksandar R. Zivkovic

Academic Editor

PLOS ONE

Journal Requirements:

2. In the ethics statement in the manuscript and in the online submission form, please provide additional information about the patient records/samples used in your retrospective study, including: a) whether all data were fully anonymized before you accessed them; b) the date range (month and year) during which patients' medical records/samples were accessed.

3. In the manuscript Methods, please 1) provide further details about how the opt-out consent mechanism and 2) confirm whether the IRB approved the opt-out consent mechanism.

Reviewers' comments:

Reviewer's Responses to Questions

**Comments to the Author**

1. Is the manuscript technically sound, and do the data support the conclusions?

Reviewer #1: Partly

Reviewer #2: Partly

2. Has the statistical analysis been performed appropriately and rigorously? 

Reviewer #1: No

Reviewer #2: Yes

3. Have the authors made all data underlying the findings in their manuscript fully available?

Reviewer #1: Yes

Reviewer #2: Yes

4. Is the manuscript presented in an intelligible fashion and written in standard English?

Reviewer #1: Yes

Reviewer #2: Yes

5. Review Comments to the Author

Reviewer #1: The study assessed the SpO2/FiO2 ratio in ARDS.

Although the topic remains of interest, the manuscript should be highly modified.

First, the authors included only 94 patients within 10 years. The number of patients is very low and the study could not respond to the question.

Second, one third of patients had interstitail lung disease. In that setting diagnosis of ARDS is very difficult and most of those patients does not have ARDS.

3- the multivariate analysis is very strange including the same variable several time (APACHE2, SOFA and SAPS are related variables).

4-comparison of AUC were not adequat : the authors should use likelihood ratio test.

5- the authors demonstrated that P/F ratio and S/F ratio are correlate. However, P/F ratio is a well known prognosis factor during ARDS. However, the authors wrote that prognostic factors for these patients that are avalaible within the first 24 hours of admission have not been fully studied (in the abstract).

Reviewer #2: My main concern is about the novelty of the results presented. Indeed, it has been previously shown, in a larger population of ARDS patients that spo2/fiO2 predicts outcomes (PMID: 26271028). Therefore, why the authors think that their results are innovative and merits publication? This is an issue that should be clearly stated in the manuscript. In other words, it is not clear what these results add to the current knowledge.

The part regarding the patients included should be moved to the results section (page 7, last paragraph).

It is unicentric and retrospective with low sample size.

Tables and Figures are included in the middle of the results section and makes the reading difficult to follow.

In the first sentence of the discussion the authors said that “Our findings underscored that S/F ratio was an easy to obtain a superior prognostic 216 factor for AHRF with bilateral opacities in the ICU.”. I would suggest saying “mehcanically ventilated patients with ARDS”. All patients included have bilateral infiltrates, were mechanically ventilated and have PF ratio <300.

6. PLOS authors have the option to publish the peer review history of their article (what does this mean?). If published, this will include your full peer review and any attached files.

Reviewer #1: No

Reviewer #2: No

---

## [Author Response · Author response to Decision Letter 0]

3 Jan 2021

Dear Reviewers

We noted our comments to the reviewer.

Reviewer #1

(Q1) The study assessed the SpO2/FiO2 ratio in ARDS. Although the topic remains of interest, the manuscript should be highly modified. First, the authors included only 94 patients within 10 years. The number of patients is very low and the study could not respond to the question.

(A1) As you point out, this is a single-center, retrospective study, and we agreed that the number of patients was insufficient to answer the clinical question. We added this fact in the manuscript as one of the limitations (Page 20, Line 296−299).

(Q2) Second, one third of patients had interstitail lung disease. In that setting diagnosis of ARDS is very difficult and most of those patients does not have ARDS.

(A2) We agree with your opinion. In this study, the target condition was an acute hypoxemic respiratory failure (AHRF) that was the heterogenous population including acute exacerbations in ILD and ARDS. Because it is difficult to make a definite diagnosis of ARDS at the time of admission to the ICU (PMID: 26903337), we focued on patients with AHRF. We believe that it is clinically useful to determine the prognostic markers for AHRF and this is one novel point in this study. We described this point in the introduction of original manuscript (Page 4 Line 67−Page 5 Line 72). In addition, we added a sentence in the discussion to emphasize this point (Page 18, Line 263−266). 

(Q3) The multivariate analysis is very strange including the same variable several time (APACHE2, SOFA and SAPS are related variables).

(A3) Thank you for appropriate suggestion. We also thought that the types and number of factors used in the multivariate logistic regression analysis were extraordinary, so we reanalyzed the results by including three factors in Table 3: age, S/F ratio, and APACHE II (Page 14 Line 196−204).

(Q4) Comparison of AUC were not adequat : the authors should use likelihood ratio test.

(A4) We agree with your suggestion. We conducted a likelihood ratio test. The goodness of fit for each model was as follows: The S/F ratio for ICU mortality in all patients, p = 0.002; The S/F ratio for hospital mortality in all patients, p = 0.002; The S/F ratio for ICU mortality in patients with bacterial pneumonia, p = 0.001; The S/F ratio for hospital mortality in patients with bacterial pneumonia, p = 0.046; The S/F ratio for ICU mortality in patients with interstitial lung disease, p = 0.113; The S/F ratio for hospital mortality in patients with interstitial lung disease, p = 0.018 (Page 14 Line 206−Page 15 Line 217).

(Q5) The authors demonstrated that P/F ratio and S/F ratio are correlate. However, P/F ratio is a well known prognosis factor during ARDS. However, the authors wrote that prognostic factors for these patients that are avalaible within the first 24 hours of admission have not been fully studied (in the abstract).

(A5) Thank you for youre precise suggestion. We rewrote the sentences in the intoroduction section of the abstract that you pointed out. Moreover, We added a description in the methods section of the abstract (Page 2, Line 31−33; Page 3, Line 37). 

Reviewer #2: 

(Q1) My main concern is about the novelty of the results presented. Indeed, it has been previously shown, in a larger population of ARDS patients that spo2/fiO2 predicts outcomes (PMID: 26271028). Therefore, why the authors think that their results are innovative and merits publication? This is an issue that should be clearly stated in the manuscript. In other words, it is not clear what these results add to the current knowledge.

(A1) Thank you for the appropriate suggestion and the article you presented. We believe there are two novelties in our study. First, our study populations are slightly different from previous studies because we included a group of patients with acute exacerbation of interstitial lung disease and ARDS as diseases causing acute hypoxic respiratory failure. Because it is difficult to make a definite diagnosis of ARDS at the time of admission to the ICU (PMID: 26903337), we focued on patients with AHRF. We believe that it is clinically useful to determine the prognostic markers for AHRF and this is one novel point in this study. Second, not many direct comparisons of the S/F ratio with multiple scoring systems (APACHE II, SAPS, SOFA), which proved useful as prognostic factors, have been performed. By conducting this comparison, we believe that we highlighted the simplicity and usefulness of the S/F ratio. However, we thought it was not easy to grasp the study's novelty in the current text as you pointed out. We reworded the first sentence of the discussion section and conclusion section (Page 16, Line 240−241; Page 21, Line 313).

(Q2) The part regarding the patients included should be moved to the results section (page 7, last paragraph).

(A2) Thank you for your valuable feedback. As per your suggestion, we moved the relevant part to the result section (Page 9, Line 148−Page10, Line 159) .

(Q3) It is unicentric and retrospective with low sample size.

(A3) We appreciate reviewer’s comments. As you mentiond, this retrospective study was conducted in a single center, and we thought that the number of patients was insufficient to answer the clinical questions. We added this point to the manuscript as one of the limitations (Page 20, Line 296−299).

(Q4) Tables and Figures are included in the middle of the results section and makes the reading difficult to follow.

(A4) Thank you for your suggestion. We inserted the Tables in the middle of the manuscript in accordance with the following PLOS ONE submission rules ”Place each table in your manuscript file directly after the paragraph in which it is first cited (read order). Do not submit tables in a separate file(s).” We also contacted the PLOS ONE office and were instructed to submit the manuscript in its current format. We apologize for any inconvenience this may cause and thank you for your cooperation. 

(Q5) In the first sentence of the discussion the authors said that “Our findings underscored that S/F ratio was an easy to obtain a superior prognostic factor for AHRF with bilateral opacities in the ICU.”. I would suggest saying “mehcanically ventilated patients with ARDS”. All patients included have bilateral infiltrates, were mechanically ventilated and have PF ratio <300.

(A5) Thank you for your valuable feedback. While our study included ARDS patients, patients with acute exacerbation of interstitial lung disease were also selected. We did not consider these patients as ARDS. Moreover, we thought that it was difficult to determine whether a patient had ARDS or not at the time of ICU admission (PMID: 26903337). Therefore, we defined the patients who had bilateral infiltrates, were mechanically ventilated, and had a P/F ratio <300 as "acute hypoxemic respiratory failure." In light of this, we revised the sentences in discussion section you pointed out as follows: Our findings underscored that S/F ratio was easy to obtain a superior prognostic factor for mechanically ventilated AHRF with bilateral opacities in the ICU to APACHE II, SAPS II, and SOFA (Page 16, Line 240−241).

---

## [Editor Report · Decision Letter 1]

11 Jan 2021

Utility of SpO2/FiO2 ratio for acute hypoxemic respiratory failure with bilateral opacities in the ICU

PONE-D-20-35947R1

Dear Dr. Yosuke,

We’re pleased to inform you that your manuscript has been judged scientifically suitable for publication and will be formally accepted for publication once it meets all outstanding technical requirements.

Kind regards,

Aleksandar R. Zivkovic

Academic Editor

PLOS ONE
---

## [Editor Report · Acceptance letter]

13 Jan 2021

PONE-D-20-35947R1 

Utility of SpO_2_/FiO_2_ ratio for acute hypoxemic respiratory failure with bilateral opacities in the ICU 

Dear Dr. Fukuda:

I'm pleased to inform you that your manuscript has been deemed suitable for publication in PLOS ONE. Congratulations! Your manuscript is now with our production department. 

Kind regards, 

on behalf of

Dr. Aleksandar R. Zivkovic 

Academic Editor

PLOS ONE